# Polymeric Nanoparticle Associated with Ceftriaxone and Extract of Schinopsis Brasiliensis Engler against *Multiresistant Enterobacteria*

**DOI:** 10.3390/pharmaceutics12080695

**Published:** 2020-07-23

**Authors:** Maísa Soares de Oliveira, João Augusto Oshiro-Junior, Mariana Rillo Sato, Marta Maria Conceição, Ana Cláudia Dantas Medeiros

**Affiliations:** 1Laboratório de Desenvolvimento e Ensaios de Medicamentos, Centro de Ciências Biológicas e da Saúde, Universidade Estadual da Paraíba, R. Baraúnas, 351, Cidade Universitária, 58429-500 Campina Grande, Paraíba, Brasil; maisasoaresdo@gmail.com; 2Faculdade de Ciências Farmacêuticas, Universidade Estadual Paulista Júlio de Mesquita Filho, Araraquara-Jaú, 14.800-903 Km 1 Araraquara, São Paulo, Brazil; rillosato@gmail.com; 3Centro de Tecnologia e Desenvolvimento Regional, Universidade Federal da Paraíba, Av. dos Escoteiros, s/n, Mangabeira VII, 58055-000 João Pessoa, Paraíba, Brasil; marta.conceicao@academico.ufpb.br

**Keywords:** bacterial resistance, Enterobacteriaceae, chitosan, HPMC

## Abstract

Bacterial resistance has become an important public health problem. Bacteria have been acquiring mechanisms to resist the action of antimicrobial active pharmaceutical ingredients (API). Based on this, a promising alternative is the use of nanotechnology, since when the systems are presented in nanometric size, there is an increase in the interaction and concentration of the action at the target site improving the activity. Thus, this study aims to develop a polymeric nanoparticle (PN) composed of chitosan and hydroxypropylmethylcellulose, as an innovative strategy for the administration of an association between ceftriaxone and extract of *S. brasiliensis*, for the treatment of Enterobacteriaceae. From a Box–Behnken design, nanoparticles were obtained and evaluated using the DLS technique, obtaining the particle size between 440 and 1660 nm, IPD from 0.42 to 0.92, and positive charges. Morphological characteristics of PN by SEM revealed spherical morphology and sizes similar to DLS. Infrared spectroscopy showed no chemical interaction between the components of the formulation. The broth microdilution technique evaluated their antimicrobial activity, and a considerable improvement in the activity of the extract and the API compared to the free compounds was found, reaching an improvement of 133 times in the minimum inhibitory activity CRO.

## 1. Introduction

Bacterial resistance to available antibiotics causes about 700 thousand deaths per year, according to data obtained by the World Health Organization (WHO), however, it is estimated that this number will increase considerably, reaching 10 million deaths per year by 2050, which makes it a major public health problem [1]. Besides, this resistance will have economic implications because studies show the world may lose between 60 and 100 trillion dollars in economic production, which represents a decrease of 2.0 to 3.5% of global gross domestic product (GDP) expected for 2050 [2,3].

The adaptive capacity of bacteria through the development of resistance mechanisms allows them to remain in activity even when in contact with antibiotic substances and this happens because bacteria can regulate the expression of genes and determinants of resistance, preventing the active pharmaceutical ingredients (API) to act effectively [4].

The vast majority of bacterial species can express resistance genes, however, the Enterobacteriaceae family is considered a special treat, since is a family that has a large number of genera and species, among them the most commonly isolated in clinical cultures, including *Escherichia coli, Klebsiella pneumoniae,* and *Enterobacter aerogenes*, often relating to infections acquired in the Community and the hospital environment [5].

The Enterobacteriaceae is a family of Gram-negative, facultative anaerobic bacteria that do not form spores, they leave a wide range of carbohydrates, have a complex antigenic structure, and produce a variety of toxins and other virulence factors. Some enteric organisms are part of the normal microbiota and eventually cause disease, but others are regularly pathogenic to humans [6,7].

Infections caused by bacteria belonging to this family are mainly treated with antibiotics belonging to the beta-lactam group. This group includes penicillins, cephalosporins, monobactams, and carbapenems which are the most frequently prescribed antibiotics worldwide. These agents bind to and inhibit bacterial enzymes (referred to as penicillin-binding proteins) responsible for cell wall synthesis. However, enterobacteria have the ability to produce beta-lactamases, which are enzymes that promote the degradation of beta-lactam rings through hydrolysis, inactivating the antibiotics of this class [8].

The β-Lactamases differ from each other in their substrate profiles (the different types of β-lactam antibiotics they inactivate), inhibitor profile (which compounds inactivate them), and sequence homology (amino acid composition of these enzymes). The extended-spectrum beta-lactamases (ESBL) are capable of hydrolyzing penicillins, cephalosporins including third generation and monobactams, while carbapenemases inactivate the classes, fourth-generation cephalosporins, and carbapenems. Thus, infections caused by bacteria producing these enzymes are hardly treated due to the few therapeutic options [5,9,10].

Due to this important ability that bacteria have to acquire resistance, and the difficulty of producing new antibacterial substances drives world governments to recognize the need to seek innovative solutions to combat this public health problem [11]. Several studies using plant extracts have been conducted based on popular knowledge. Among the species studied, *S. brasiliensis* revealed numerous pharmacological and antimicrobial activity [12,13]. In addition, other studies evaluated multiresistant microorganisms and observed the modulating effect of the extract with various antibiotics, improving their response [14,15]. These numerous pharmacological proprieties may be due to the bioactive compounds present in this species, which include tannins and polyphenols, such as methyl gallate, gallic acid, and ellagic acid [16,17,18,19].

Another idea to improve the treatment of infections is the use of nanotechnology, since it is possible with nanoencapsulation to modify the activity of antibiotics already available in the market that had their action decreased due to resistance mechanisms presented by bacteria [20].

When the systems are in nanometric size and the surface charges are adequate, they promote an increase in the interaction and concentration of the active in the target site, protecting the API from enzymatic degradation, in addition to improving their penetration into the mucosal epithelium, modulating the pharmacokinetics of API and thus improving their effectiveness and reducing toxicity [21,22,23,24,25,26,27].

Based on this, the polymeric nanoparticles (PNs) have become an important alternative against bacterial resistance, since they can recover the effectiveness of drugs to which bacteria already have resistance, overcoming mechanisms such as degradation by beta-lactamase, efflux pumps or difficulty to cross broad cell walls. Additionally, PNs are less likely to induce resistance when compared to free active pharmaceutical ingredients(API) [28].

Therefore, this study aimed at developing and characterizing a polymeric nanoparticle associated with ceftriaxone and extract of leaves of *S. brasiliensis*, for the treatment of enterobacteria producing ESBL and KPC.

## 2. Materials and Methods

### 2.1. Materials

Chitosan (LMWC) and colloidal silicon dioxide (MW = 60.08 g mol^−1^) were purchased from Henrifarma Produtos Químicos e Farmacêuticos Ltd. (Cambuci, Brazil). Hydroxypropyl methylcellulose (MW = 1261.45 g mol^−1^) from Unna Derme Comércio de Produtos Farmacêuticos Ltd. (Campina Grande, Brazil).

### 2.2. Plant Material

The leaves of *Schinopsis brasiliensis* Engler were collected in the semiarid region of Paraíba, Brazil (70°13′50′ S, 35°52′52′ W). The Macro and microscopic botany identification were performed in the Herbarium Jaime Coelho de Moraes in the Center of Agricultural Sciences of the federal university of Paraíba (UFPB) and the exsiccate was deposited with registration number EAN—14049. The access to the plant was registered in the National System of Genetic Heritage and Associated Traditional Knowledge (SISGEN) under the number A0AAB55.

### 2.3. Obtaining the of S. brasiliensis Extracts

To obtain the extracts, the plant drug was cleaned and dried in a drying oven by forced air circulation at 40 °C, until constant weight and pulverized in a knife mill with an output of 10 mesh. After the processing, the plant drug was stored in Kraft paper bags. Dry residue and extraction yield was determined by analysis, as described in the Brazilian Pharmacopeia 5th Edition [29].

The hydroalcoholic extract was obtained by the maceration method, using as solvent a hydroalcoholic solution 70% (v/v). After this step, the extract was dried in a spray dryer using as pharmaceutical stabilizer aerosol 200^®^ 20%, which was calculated concerning the dry residue. For the nebulization of the extract was used 120 °C as input temperature and between 90 and 95 °C as output temperature, with a feed flow of 7 mL min^−1^.

The dried extracts were stored in hermetically sealed vials under a temperature of 20 °C until further analysis.

### 2.4. Development of Polymeric Nanoparticles

Polymeric nanoparticles (PN) were developed according to the polyelectrolytic complexation methodology, using a cationic charge polymer (chitosan (CS)) and an anionic charge polymer (HPMC) according to Boni et al. [30], CS was dispersed in 0.1 mol L^−1^ acetic acid, while HPMC was dispersed in distilled water obtaining in both, a final concentration of 2 mg mL^−1^. The pH of the solutions were obtained with the aid of a pHmeter Mylabor (OA210A) brand and the adjustment was performed by dripping an alkaline solution of NaOH 1M, to obtain a final pH of 5.5. To eliminate possible agglomerates of particles, a polymeric solution was filtered using 0.25 μm filters Millipore brand.

Then, the PNs were obtained by placing the API and the dried extract in the dispersion of the polymers with great volume (Table 1), followed by the slow addition, with the aid of a 12.7 mm syringe and needle, of the polymer dispersion of opposite charge under magnetic stirring for 15 min, at room temperature (25 °C). 

### 2.5. Experimental Design Box–Behnken

The optimization of the PNs formulation process was performed through an experimental design of Box–Behnken type using three independent variables: concentration of polymers (X1), API concentration (X2) and extract concentration (X3), which were established at high, medium, and low levels (Table 2), totaling 12 runs with three repetitions at the midpoint to obtain the dependent variable (response): particle size. The Statistica 10 software (StatSoft, version 10.0.228.8) was used to generate the planning matrix.

### 2.6. Physicochemical Characterization of Nanoparticles

#### 2.6.1. Analysis of Particle Size, Polydispersity Index (PDI) and Zeta Potential (ZP)

To determine the particle size and the PDI was used dynamic light scattering technique (DLS) and to determine the ZP has used the electrophoretic mobility technique, both with the aid of Zetasizer equipment Nano-ZS at 25 °C, at a detection angle of 173°. The nanoparticles were placed in the sample port and the readings were made in triplicate.

#### 2.6.2. Morphological Analysis of Nanoparticles

Morphological analysis of nanoparticles was performed by high-resolution scanning electron microscopy (SEM-FEG) technique, model JSM-7500F (JEOL Brazil, São Paulo, SP, Brazil), with operating software PC-SEM with secondary electron detectors, backscattering and chemical analysis (energy dispersive spectroscopy (EDS)) Thermo Scientific, model Ultra Dry, with operating software NSS 2.3.

The samples were diluted in the ratio 1:50 (v/v) in Tween 20 (0.5% v/v) to avoid agglomeration of particles, and 7 µL added on a silicon metal plate of 1cm^2^ mand taken to desiccator, where they remained for three days at room temperature. Subsequently, the support containing the sample was coated with carbon using the equipment BAL-TEC SCD 050 (sputter coater), with associations of mechanical pumps and turbo brand Edwards model t-station 75, for reading the analysis.

#### 2.6.3. Fourier Transform Infrared Spectroscopy (FTIS)

The tests were performed with a Shimadzu spectrophotometer IR prestige-21 model, in the region of 4000–400 cm^−1^. The resolution of analysis was 4 cm^−1^ and the samples were dispersed in KBr (degree of spectroscopic purity) in the ratio 1:100 mg for making tablets of 1.2 cm in diameter.

### 2.7. Antimicrobial Activity 

The determination of the Minimum Inhibitory Concentration (MIC) of the extract of *S. brasiliensis* (EX) and ceftriaxone (CRO) was performed against strains of enterobacteria with different sensitivity profiles. For this, we used a strain of *Escherichia coli* ATCC 25,922 (B01) and clinical strains from the collection of a private laboratory network of Campina Grande, with known sensitivity profile, being an *Escherichia coli* producing ESBL (B02) and a carbapenemase-producing *Klebsiella pneumoniae* (KPC) (B03). To determine the MIC was used the broth microdilution technique, as described by the Clinical Laboratory and Standards Institute.

To perform the test, the inoculum was standardized in a spectrophotometer at a wavelength of 625 nm to obtain the correct optical density of turbidity control, which must vary from 0.08 to 0.10 to obtain the standard McFarland solution of 0.5, resulting in a suspension containing approximately 1 to 2 × 10^8^ CFU mL^−1^. To obtain a final concentration of 5 × 10^6^ CFU mL^−1^ a dilution of 1:20 was performed in the suspension obtained. 

The extracts were diluted in 10% dimethylsulfoxide (DMSO) at concentrations of 1000 to 7.8 μg mL^−1^, while a solution of ceftriaxone was obtained and diluted from 1000 to 0.48 μg mL^−1^. One hundred microliters of each extract and ceftriaxone were added in the wells of the microplate, containing the broth 80 μL Mueller Hinton and 20 μL of the adjusted inoculum, resulting in a final concentration 5 × 10^5^ CFU mL^−1^. The plates were incubated at 35 ± 2 °C for 20 h.

The MIC was defined as the lowest concentration that inhibited visible microbial growth, confirmed after the addition of 20μL of resazurin in each well of the plate. The analyses were performed in triplicate. In parallel, the viability of the strain (growth control) and the sterility control of the medium was performed.

The determination of the MIC of PN was obtained following the methodology described in topic 4.3. Pure nanoparticles, nanoparticles containing only 3% API, nanoparticles containing 3% extract, and the formulation chosen according to the results obtained by characterization were tested. From the results obtained by the MIC, it was determined the minimum bactericidal concentration (MBC). With the aid of the platinum loop, 1µL was removed from the well and seeded in Petri dishes with Mueller Hinton agar. The plates were incubated at 35 ± 2 °C for 24 h. After this, it was possible to observe whether there was bacterial growth.

## 3. Results and Discussion

### 3.1. Evaluation of S. brasiliensis Extract

The dry residue of the extract was determined according to what is described in the Brazilian Pharmacopoeia V Edition (2010) [29] being obtained 64 g of 200 mL of the macerated extract. 12.8 g of the aerosol 200^®^ stabilizer were added, which represents 20% of the dry residue mass, resulting in a theoretical mass of 76.8 g. After drying the extract containing the stabilizer, a mass of 67.0 g of the dry extract was obtained, which represented an extraction yield of 87.2%.

### 3.2. Development of Nanoparticles

The polyelectrolytic complexation was the chosen technique, since it has advantages, such as the ease of two polymers of opposite charges complex without the need for reaction initiators, catalysts, or crosslinking. The elimination of these additives makes most complexes nontoxic and easy to manufacture, and this reduces the cost of research and development of drug compounds [31].

Thus, it was determined that the optimization of obtaining the nanoparticles would be done using the experimental design of the Box–Behnken type, since experimental designs are useful for the development of formulations, because from them less a smaller amount of experiments are needed in relation to pharmacotechnical development empirically, and provide information that correlates the independent and dependent variables [32].

The combinations between the independent variables (HPMC/CS, CRO, and EX) generated the planning matrix and from this, 15 formulations were prepared, three of them repetitions at the central point. Table 1 shows the composition of the formulations based on the independent variables. According to the planning matrix generated from the software Statistic 10, the PN were prepared by varying the concentrations of the polymers so that the upper level had a predominance of CS (70%) and the complete volume with HPMC (30%), the low level, the inverse, and the average level equal amounts of the two polymers. 

The amount of API and extract added in the PN was equivalent to 3%, 2%, and 1% of the total polymer value in the preparations. Figure 1 shows the visual aspects of all formulations developed according to the planning matrix.

The obtained nanoparticles showed homogeneity, revealing that it was possible to incorporate 3% (w/w) of ceftriaxone and extract with the polymer mass without observing the presence of precipitates or agglomerates. However, due to the formulations present extract in its composition, it was possible to observe a slight change in the shade from transparent to light green, which is due to the color of the extract of the leaves of *S. brasiliensis* in the meantime, where transparency has been maintained. Visually, no formulation showed unwanted organoleptic characteristics. 

Using the polyelectrolytic complexation technique to obtain nanoparticles containing curcumin, Tan et al. [33] obtained similar results. Initially, it was prepared with a chitosan solution and another with curcumin that was dispersed in the first, followed by drip Arabic gum solution, managing to incorporate 4% (m/m) of the bioactive compound curcumin. 

Thus, subsequently, these formulations followed for physicochemical characterization to obtain answers for processing the Box–Behnken design and selection of the formulation for the sequence of experiments.

### 3.3. Dynamic Light Scattering, Zeta Potential, and Experimental Design

The dynamic light scattering (DLS) technique is often used to determine the size of PN. The colloidal suspension is illuminated by a monochromatic laser light that is scattered in a photon detector. Due to the particles present Brownian motion, the intensity of the scattered light detected fluctuates in time and this is related to the size of the particles [34,35]. The results of particle size, PDI and zeta potential are shown in Table 3.

The particle size values obtained were placed in the planning matrix generated by the software (Statistical 10.0) obtaining then the Pareto diagram with the estimated effects of the variables tested about the size of PN (Figure 2). The result showed that only the quadratic portion of polymer in the upper level interfered in the size of the particles tested, which indicates that the increase in the concentration of chitosan in the formulation influences the particle size. However, it was observed a lack of adjustment in the model. Thus, the model is not suitable to generate the response surface graphs with accuracy. 

However, these results corroborate the data found by dynamic light scattering technique (Table 3), in which it is possible to verify that the chitosan in the proportion of 30% presents average diameter values (d.nm) between 440 ± 2.1 and 497 ± 1.9 nm. While the formulations contained chitosan in the proportion of 50% and 70%, the values are greater than 500 nm. In agreement with the results found, studies conducted by Zaki and Hafez [36], observed larger PN sizes related to the increase in the proportion of chitosan, reaching a variation of 152 nm, since the ratio 5/1 of CS/TPP at pH 4 the average particle size was 128 nm which was increasing the measure that increased the proportion of chitosan reaching 280 nm. 

The formulations named as PN12, PN13, PN14 and PN15 the values of d. nm were 1094 ± 8, 1430 ± 7, 1660 ± 10 and 1275 ± 5, respectively, exceeding the nanoscale. Studies by Polexe and Delair [37] with nanoparticles, aiming at the functionalization of antibodies obtained by polyelectrolytic complexation technique using the polymer chitosan and hyaluronic acid showed similar results to those presented in this study, since the particle size ranged between 271 and 1220 nm. Thus, the authors attributed the fact that some particles have a larger size (1100 and 1220 nm) by the proportion of polymers used, which led to a neutralization of the charges preventing the formation of the complex.

This hypothesis is consistent for the formulations PN12, PN13, PN14, and PN15, since it was observed that in these formulations obtained with the same proportion of CS/HPMC, a considerable increase in particle size, which may have been due to charge cancellation and absence of PN formation. Moreover, these formulations were discarded for the other experiments.

To obtain PN by polyelectrolytic complexation, Boni et al. [30] used the positive charge polymers chitosan and the negative charge, hyaluronic acid, and HPMC. The PN obtained with and without API showed sizes ranging between 325.7 and 450.5, and zeta potential ranging between ±20.9 and ±33.1, which is similar to the results found in the formulations named PN-1, PN-2, PN-3, and PN-4.

In the literature, different studies attribute to the smaller particle size the ability to cross the biological barrier, improving the absorption of API, when there is a reduction in this size. This fact represents numerous advantages, because it improves the bioavailability of the drug, and the length of stay in the infected site, protecting the drug from degradation and achieving a gradual release pattern [38].

Regarding the results of PDI, we can observe that the formulations developed with the lowest concentration of chitosan have fewer variations in the results regardless of the concentration of the mixture API/extract, being between 0.42 ± 4.13 and 0.58 ± 10. Besides, they have smaller relative deviations when compared to formulations developed with higher concentrations of chitosan (0.47 ± 5.49 and 0.92 ± 9.05), indicating less variation in particle size distribution.

According to Avadi et al. [39], the values of IPD vary between 0 and 1, and values less than 0.5 indicate that the particles have a homogeneous distribution, the particles have a size that does not vary so much about the average, while values above this indicate a more heterogeneous distribution. However, according to current studies conducted by Danaei et al. [40] the values of this index found above 0.7 indicate that the sample has a very wide particle size distribution and is probably not suitable to be analyzed by DLS technique.

The importance of obtaining PN with monomodal distribution is due to its physicochemical properties, since the absence of this distribution can affect the volume properties, product performance, processability, stability and appearance of the final product, and influence cell uptake dependent on endocytosis [40]. The increase of the IPD, which indicates a heterogeneous distribution may be due to the presence of aggregates, however, these results should not be analyzed in isolation and it is necessary to combine with other techniques [41].

The ZP values found for all formulations were positive being in a range between 18.15 ± 11.2 and 38.95 ± 5.99. The positive charges found are due to free amino groups (–NH2) of chitosan that become protonated (–NH_3_^+)^ (pH 5.5) overcoming the negative groups present in HPMC (–OH) [42,43]. 

This effect may be related to the absorption of anionic groups by the long amino groups of chitosan, keeping high the value of the electrical double layer thickness, which, in turn, prevents aggregation. Because of this, the ZP is one of the most used parameters to indicate long-term stability, due to repulsion between particles (electrostatic forces) [39]. 

According to Bhattacharjee [44], the dispersions of np that have ZP in the range of ±20–30 mV are considered moderately stable, which allows them to be used in drug administration. Most of the values found in this study are within this range, which allows this classification. Additionally, a study by Ilk et al. [45] evaluated the stability of chitosan np loaded with kaempferol and obtained a ZP range very close to that found in this study, ranging between ±18.5 and ±38.1, which observed adequate stability of the product, which was maintained for 30 days.

Based on the results were selected to continue the analysis PN-1, PN-2, PN-3, and PN-4 formulations since they obtained the smallest particle sizes combined with appropriate ZP and PDI.

### 3.4. Scanning Electron Microscopy (SEM)

The chosen formulations were evaluated in SEM. This technique can provide information about the size and morphology of the particles. An electron beam falls on the sample surface to produce a variety of signals that are collected by a detector, resulting in images with high magnification (50–10,000×) and resolution from 10 nm to micrometers [46]. Figure 3 shows the photomicrographs of the nanoparticles.

The SEM results show that PN-1 (3a), PN-2 (3b), PN-3 (3c), and PN-4 (3d) have spherical morphology with nanometric sizes between 150 and 500 nm, similar to the sizes found in the DLS particle distribution graph. Due to the lack of contrast between the active and inert components used, it was not possible to distinguish differences between them.

Probably, the API and the extract are trapped inside the PN and/or interacting on the surface by hydrogen bonds. Corroborating what was previously discussed regarding the need for a combination of techniques to evaluate the homogeneity of the sample, from the result obtained in the SEM, it is possible to observe that there was no great difference between the sizes of the PN obtained, which differs from the results obtained in the PDI. 

The studies of Gaumet et al. [47] compared results obtained in an electronic micromicron with the PDI values. At microscopy, the sizes found ranged between 100 nm and 1 nm, while by light scattering technique, the average size was 318 nm with PDI of 0.093 which is considered a low value that would represent monomodal distribution. Based on this information it is important to highlight the need for a combination of techniques to evaluate the morphology of the particles since due to chitosan by a cationic polysaccharide their physicochemical properties vary according to pH.

### 3.5. Fourier Transform Infrared Spectroscopy (FTIR)

FTIR is a vibrational surface chemical analytical technique that measures the intensity of infrared versus the wavelength of light. This technique is used to chemical characterization of materials at the molecular level, since it determines the positions and relative intensities of all absorptions, or peaks, in the infrared region and graphically records them [48,49].

Figure 4 shows the FTIR spectra of the excipients used in the formulation separately, solution of CS (Figure 4a), and HPMC (Figure 4b).

In the CS spectrum, it was possible to observe a wide absorption peak in the region of 3390 cm^−1^ related to the stretching of the -OH bond, and there may be overlap in the NH stretch band that occurs between 3500 to 3100 cm^−1^ [50]. Besides, there is a peak at 2879 cm^−1^ which corresponds to the stretching of the C–H bond of chitosan, and two other peaks at 1652 and 1598 cm^−1^ which are attributed to primary and secondary amide respectively [51]. Another main band was found ay 1095 cm^−1^ referring to the C–O stretching bond [52].

Analyzing the spectrum of HPMC is possible to observe absorption bands characteristic of this polymer as described in the literature [53]. In the region of 3481 cm^−1^ was generated broadband that corresponds to the O–H connection, while at 2893 cm^−1^ there is the formation of a peak stretch of C–H sp^3^ and at 1384 cm^−1^ the peak obtained was related to the folding absorption of CH_3_. Another peak observed was at 1058 cm^−1^ which is characteristic of the C–O strain [54].

All bands observed in Figure 4 are typical and similar to those described in the literature mentioned above and are present in all commercial samples, revealing that all have the same functional groups.

The spectrum of ceftriaxone (Figure 5a) shows two absorption bands at 3442 and 3253 cm^−1^ referring to the stretching of the N–H group in amides and another main peak at 1735 cm^−1^ which is due to the stretching of the C=O bond of cyclic amide (lactam). At 1652 cm^−1^ there is a peak that the literature assigns to the oxime that generates the absorption C=N and an O–H absorption between 3650 and 2600 cm^−1^ that is not easily observed due to the overlap of bands. Another peak generated at 1033 cm^−1^ refers to the C–O stretch of the ether [55].

Figure 5b represents the spectrum of the extract of *S. brasiliensis* which has a wide band at 3334 cm^−1^, which refers to the stretch of the O–H bond. Its presence is justified because the extract is composed of several secondary metabolites, including polyphenols, which are rich in hydroxyl groups [18]. Additionally, at 1718 cm^−1^ was observed a peak that was attributed to the strain of the bond C=O and another related to the bond C=C at 1606 cm^−1^. It is possible to observe two peaks at 1207 and 1105 cm^−1^ that appear due to the presence of the stretching of the C–O bond of the ester, a compound present in flavonoids, such as aglycone, found in this extract [56]. 

In addition to the spectra of the active and inert components used in the formulation alone, spectra of PN were obtained with and without the extract and the API, to verify whether there was chemical interaction between the components (Figure 6).

From the analysis of the spectra obtained in Figure 6, it is possible to observe that the functional groups found in PN with and without extract and API, (PN-4) are similar, since there was no disappearance of the main absorption peaks, visualized in the inert excipients (Figure 4) and active (Figure 5). The FTIR spectra suggest that there was no appearance of new molecular groups and probably the molecular structure of the activity did not change.

### 3.6. Evaluation of the Antimicrobial Activity of S. brasiliensis and Ceftriaxone

The extract of *S. brasiliensis* tested against a standard strain of *Escherichia coli* by microdilution technique showed inhibitory activity up to a concentration of 250 μg, however, in the clinical strains producing resistance mechanisms previously known, the extract did not inhibit bacterially. The results obtained from ceftriaxone showed that it was able to inhibit the ATCC strain in the lowest concentration tested, however, only in the highest concentrations there was inhibition of the growth of other bacteria (Table 4).

According to Holetz et al. [57] extracts that have a MIC lower than 100 µg mL^−1^ have excellent antimicrobial activity, while those with MIC between 100 and 500 µg mL^−1^ have moderate activity and values in the range of 500 to 1000 µg mL^−1^ are considered with low activity. Extracts with MIC greater than 1000 µg mL^−1^ are considered inactive. Based on this information, it is possible to observe that the extract of *S. brasiliensis* tested showed moderate antimicrobial activity against B01, since the MIC found was 250 µg mL^−1^. However, against strains producing resistance mechanisms the extract was considered inactive (MIC > 1000 µg mL^−1^).

Similar results were found by Saraiva et al. [12] who evaluated the extract of *S. brasiliensis* against different bacterial strains, including and *E. coli* and *K. pneumoniae*. It was observed that among the six strains of *E. coli* tested, two of them found MIC of 250 µg mL^−1^, one of 500 µg mL^−1^, and three of 1000 µg mL^−1^, which led the authors to consider the activity of this extract low for this bacterial species. Regarding the strains of *K. pneumoniae*, it was found that the extract showed no activity, since the MIC found were all higher than 1000 µg mL^−1^.

Accordingly, another study by Formiga Filho et al. [13] evaluated the activity of hydroalcoholic extracts of the bark and leaf of *S. brasiliensis*. The analyzes were performed against bacterial strains of different species, including *E. coli* (ATCC 25922), and it was evaluated that the bark extract only inhibited growth at the highest concentration tested (500 mg mL^−1^), while the leaf extract showed a MIC of 200 mg mL^−1^, values above the ideal to consider the extract as active.

### 3.7. Antimicrobial Activity of Nanoparticles

Based on the results shown in this study, it was observed that the tested microorganisms that had multidrug resistance showed no sensitivity to the extract, while the MIC found in ceftriaxone was 1000 µg mL^−1^, a value far above the cut-off point (1 µg mL^−1^) to be considered sensitive [58]. Based on the characterization were tested formulations of pure PN, PN with 3% API (PNF), PN with 3% extract (PNE), and the chosen formulation (PN4) based on the characterization. The results are described in Table 5.

When testing the PN formulations the results showed that they had an important antimicrobial activity due to their ability to inhibit bacterial growth as shown in Table 5. It was possible to observe a considerable improvement in the action of encapsulated compared to free API, since the MIC found in B02 microorganism was initially 1000 μg mL^−1^, while in the chosen formulation the MIC found was 7.5 μg mL^−1^ of API, which represents an improvement of 133 times. Similarly, PN was able to inhibit B03 bacteria more effectively than API and free extract. The bacterial strain B01, which is a strain of *E. coli* sensitive to most antibiotics, was also sensitive to all tested formulations, demonstrating that PN has an important bacterial activity both for sensitive strains, as for those that have important resistance mechanisms.

From the results of the MIC, it was possible to obtain the results of MBC and it was observed that some PNS were able not only to inhibit bacterial growth but also to inactivate bacterial cells, obtaining a bactericidal activity. The results are described in Table 6.

The results show that the MBC values were similar to those found in the MIC, with a difference only in the values found in the NPS, requiring a higher concentration for the bactericidal activity to occur. However, it was observed that there was an improvement in the activity of PN when it contained in its composition the API and the extract in relation to the others. This improvement in activity can be attributed to the antimicrobial capacity that the extract of *S. brasiliensis* has, and may have been potentiated to be encapsulated in PN along with API, obtaining an improvement in its solubility when compared to the MIC found in the extract alone.

Similarly, studies by Liu et al. [59], developed polymeric nanoparticles containing poly(lactic-co-glycolic acid) (PLGA) to encapsulate the bioactive component curcumin and evaluate its antimicrobial capacity. It was observed that to have a significant reduction in *E. coli* viability required low concentrations of curcumin compared to the free bioactive and this was due to improved solubility and ability to direct curcumin bacteria.

In addition, PN was also tested, and it was possible to observe inhibitory activity as well as bactericidal activity on the three bacterial strains tested. This activity may be due to the antimicrobial properties previously described as chitosan. Several studies attribute the antimicrobial activity of chitosan to its positive surface charges, which is often reflected in ZP formulations that use this polymer in their composition [60,61].

This antimicrobial activity can happen of two main formals, one is due to the attraction of polycation particles to the negatively charged bacterial surface, which leads to disruption of bacterial membranes, causing leakage of cytoplasmic components. The other occurs because permeation of PN in the membrane can bind to intracellular components such as DNA, ribosomes, and enzymes, interrupting the normal cellular mechanism, resulting in cell death [62,63].

In this study, PN containing API showed considerably higher antibacterial activity than free API, which may be due to the ability of PN to protect the API from degradation caused by enzymes produced by the tested bacteria, improving its bioavailability. In addition, PN can control the release of loaded antimicrobial drugs, which is useful to direct the drug to its site of action [7].

In agreement with the results described here regarding the improvement of encapsulated API activity in relation to free, studies by Jamil et al. [64] evaluated the activity of chitosan PN loaded with cefazolin at concentrations of 200, 800 and 2000 mg mL^−1^, against Gram-negative bacterial strains, including an *E. coli* producing ESBL and a multidrug-resistant *K. pneumoniae.*

It was possible to observe through the agar well diffusion technique that the free cefazolin was not able to form inhibition zone, while the PN showed halos that increased in size as the API concentration in the PN increased. Similarly, the broth dilution technique showed that even at the lowest concentration of API (200 µg mL^−1^) was observed an efficient inhibition of bacterial growth compared to the free API that showed mic of 1000 µg mL^−1^ for *E. coli*, while for *K. pneumoniae* was not possible to determine since it was found higher than the tested concentration.

Furthermore, Abdelkader et al. [61] also obtained chitosan PN loaded with an antimicrobial class of beta-lactams, meropenem, and evaluated the inhibitory and bactericidal capacity of these PN compared to API on sensitive and resistant strains of *E. coli* and *K. pneumoniae*. It was observed that the dispersion of nanoparticles loaded with API had a MIC twice lower against meropenem sensitive *E. coli* and meropenem sensitive and resistant *K. pneumoniae* strains compared to free API meropenem, while the meropenem resistant *E. coli* strain was not significantly different.

## 4. Conclusions

According to the results, it is possible to conclude that the steps of development of nanoparticles, using the technique of polyelectrolytic complexation by drip, and the polymers chitosan and HPMC were suitable since from the physicochemical characterization it was found that the size of the particles obtained was within the acceptable range to be considered nanoparticles. From the antimicrobial activity assays, it was evaluated that the extract *of S. brasiliensis*, as well as ceftriaxone, showed inhibitory activity only on the strain of *E. coli* ATCC 25922, while they were not able to inhibit bacterial strains that have resistance mechanisms.

After obtaining the nanoparticles containing the extract and API it was possible to observe an important inhibitory and bactericidal activity on all bacterial strains tested, which demonstrates that nanostructured systems are able to promote improvements in the delivery of API, which make them an important alternative for the treatment of infections caused by multiresistant bacteria.

## Figures and Tables

**Figure 1 pharmaceutics-12-00695-f001:**
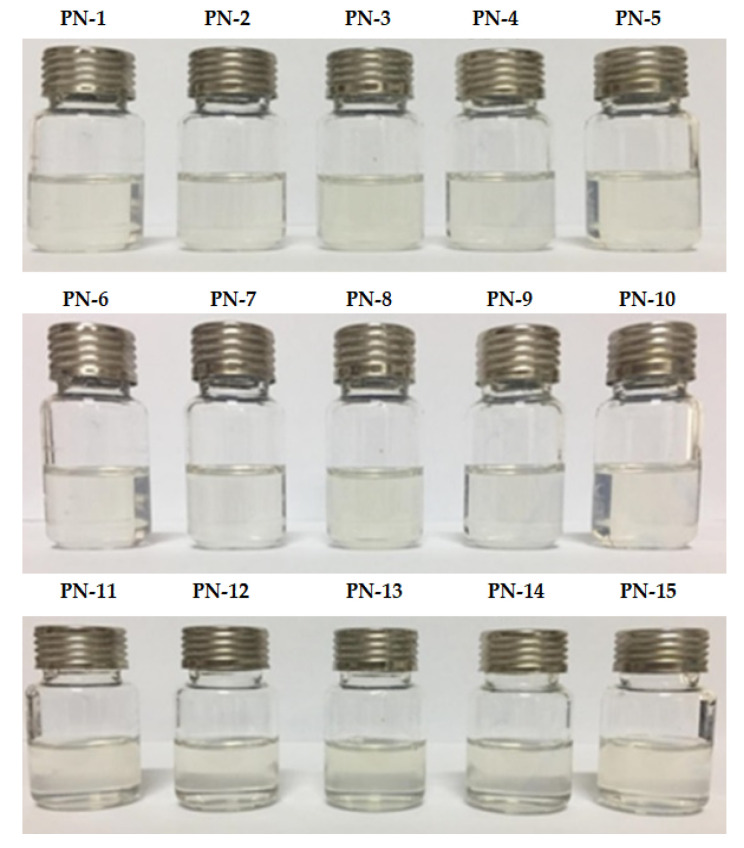
Visual characteristics of polymeric nanoparticles obtained according to the planning matrix.

**Figure 2 pharmaceutics-12-00695-f002:**
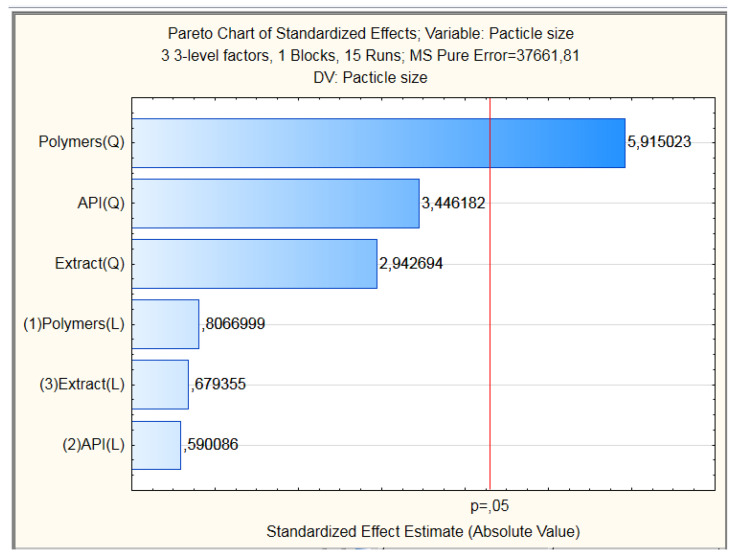
Pareto graph on the influence of the formulation components on the average size.

**Figure 3 pharmaceutics-12-00695-f003:**
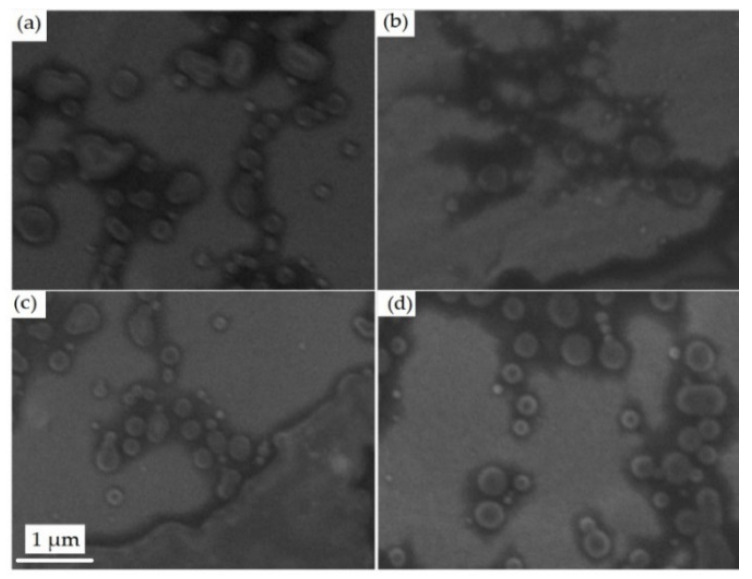
Photomicrographs of polymeric nanoparticles (**a**) PN-1, (**b**) PN-2 (**c**) PN-3, and (**d**) PN-4. Magnification of 25,000×.

**Figure 4 pharmaceutics-12-00695-f004:**
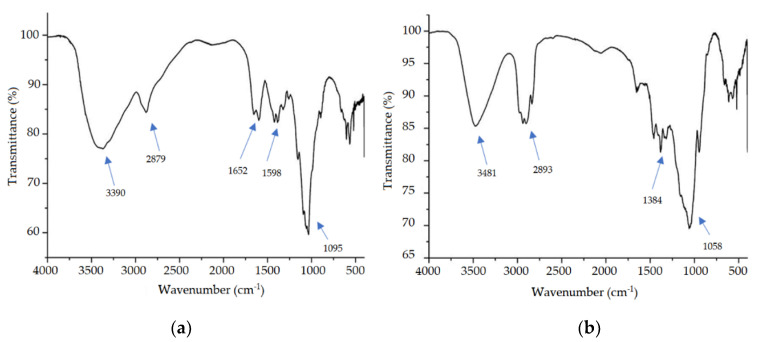
FTIR spectra for chitosan (**a**) and hydroxypropylmethylcellulose (**b**).

**Figure 5 pharmaceutics-12-00695-f005:**
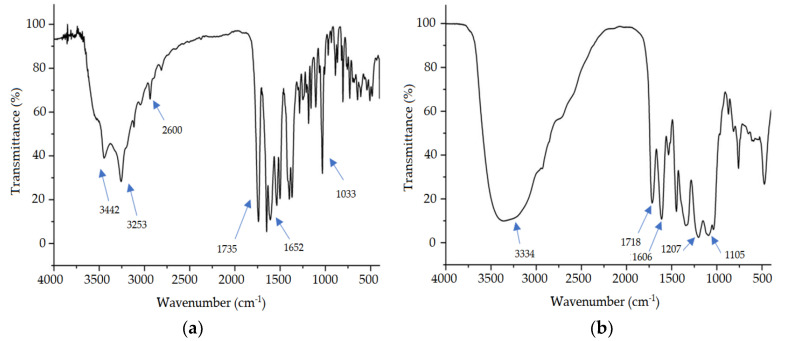
FTIR spectra for ceftriaxone (**a**) and extract of *S. brasiliensis (***b**).

**Figure 6 pharmaceutics-12-00695-f006:**
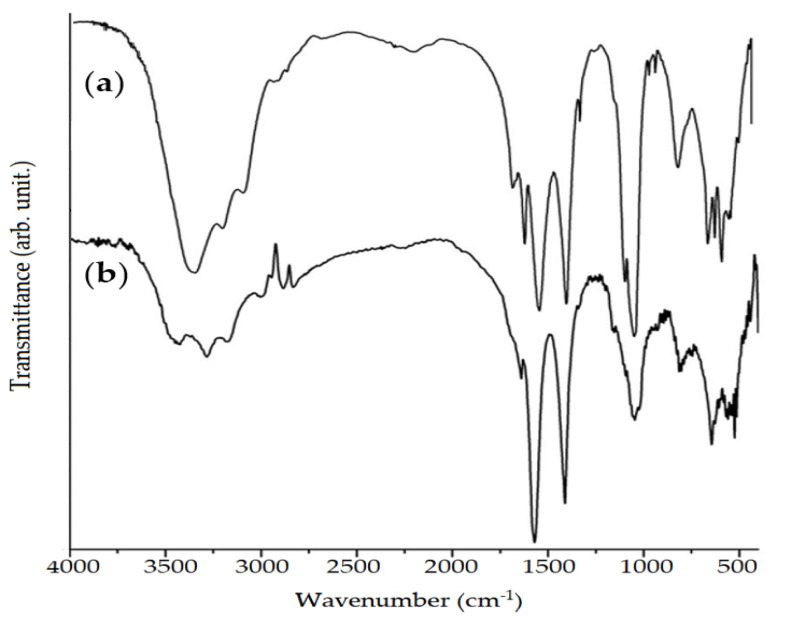
FTIR spectra of the polymeric nanoparticle (PN) (**a**) and PN containing extract and active pharmaceutical ingredients (API) (PN-4) (**b**).

**Table 1 pharmaceutics-12-00695-t001:** Composition of formulations based on the independent variables. X1—concentration of polymers (%), X2—API concentration (%), and X3—extract concentration (%).

Formulations	X1	X2	X3
	HPMC/CS	CRO	EX
	(%)	(%)	(%)
NP-1	70/30	1	2
NP-2	70/30	3	2
NP-3	70/30	2	1
NP-4	70/30	2	3
NP-5	30/70	1	2
NP-6	30/70	3	2
NP-7	30/70	2	1
NP-8	30/70	2	3
NP-9	50/50	1	1
NP-10	50/50	1	3
NP-11	50/50	3	1
NP-12	50/50	3	3
NP-13	50/50	2	2
NP-14	50/50	2	2
NP-15	50/50	2	2

**Table 2 pharmaceutics-12-00695-t002:** The Box–Behnken (3^3^) factorial and levels design using three independent variables. X1—Concentration of polymers (%), X2-API concentration (%) and X3-extract concentration (%), which were established at high, medium, and low levels.

Variables	Level		
	−1	0	1
X_1_	30/70	50/50	70/30
X_2_	1	2	3
X_3_	1	2	3

**Table 3 pharmaceutics-12-00695-t003:** Results of particle size, polydispersity index (PDI) and zeta potential for the PN measured after preparation, *n* = 3.

Acronym	Formulation	d.nm (nm)	PDI	ZP
PN1	HPMC_CS 70/30 C-B 1/2	440 ± 2.13	0.42 ± 4.13	18.15 ± 11.20
PN2	HPMC_CS 70/30 C-B 3/2	497 ± 1.93	0.58 ± 10.0	21.10 ± 0.82
PN3	HPMC_CS 70/30 C-B 2/1	494 ± 0.47	0.50 ± 2.08	22.85 ± 2.16
PN4	HPMC_CS 70/30 C-B 2/3	486 ± 0.82	0.47 ± 8.41	23.70 ± 0.59
PN5	HPMC_CS 30/70 C-B1/2	546 ± 0.38	0.69 ± 11.00	22.15 ± 2.87
PN6	HPMC_CS 30/70 C-B 3/2	554 ± 1.04	0.42 ± 6.12	22.20 ± 2.54
PN7	HPMC_CS 30/70 C-B 2/1	628 ± 1.77	0.61 ± 12.70	23.70 ± 0.59
PN8	HPMC_CS 30/70 C-B 2/3	632 ± 1.57	0.47 ± 5.49	27.50 ± 10.10
PN9	HPMC_CS 50/50 C-B 1/1	776 ± 6.92	0.42 ± 11.30	23.10 ± 3.67
PN10	HPMC_CS 50/50 C-B 1/3	713 ± 3.54	0.58 ± 18.80	22.65 ± 2.80
PN11	HPMC_CS 50/50 C-B 3/1	655 ± 14.14	0.62 ± 24.30	22.10 ± 1.91
PN12	HPMC_CS 50/50 C-B 3/3	1094 ± 8.22	0.64 ± 6.23	22.00 ± 6.42
PN13	HPMC_CS 50/50 C-B 2/2	1430 ± 7.31	0.91 ± 5.38	38.95 ± 5.99
PN14	HPMC_CS 50/50 C-B 2/2	1660 ± 10.90	0.90 ± 7.43	26.05 ± 4.07
PN15	HPMC_CS 50/50 C-B 2/2	1275 ± 5.09	0.92 ± 9.05	26.05 ± 4.07

**Table 4 pharmaceutics-12-00695-t004:** Determination of the MIC of the extract of *S. brasiliensis* and ceftriaxone against strains of Enterobacteriaceae. B01: E. coli ATCC (25922); B02: *E. coli* extended-spectrum beta-lactamases (ESBL)-producing; B03: *K. pneumoniae* that produces KPC; EX: extract of *S. brasiliensis*; CRO: ceftriaxone.

Microorganism Isolated	EX	CRO
	MIC (µg mL^−1^)	MIC (µg mL^−1^)
B01	250	<0.5
B02	>1000	1000
B03	>1000	500

**Table 5 pharmaceutics-12-00695-t005:** MIC determination of the nanoparticles obtained against microorganisms of clinical importance.

Formulations	CIM (µg mL^−1^)/Microorganisms Tested		
	B01	B02	B03
	(API/Extract)		
PNF	≤0.23	7.5	15
PNE	≤0.23	15	15
PN4	≤0.15	5	10

**Table 6 pharmaceutics-12-00695-t006:** Minimum bactericidal concentration (MBC) determination of the nanoparticles obtained against microorganisms of clinical importance.

Formulations	MBC (µg mL^−1^)/Microorganisms Tested		
	B01	B02	B03
	(API/Extract)		
PNF	≤0.23	15	15
PNE	≤0.23	15	30
PN4	≤0.15	5	10

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
