# Peer review of "Polymeric Nanoparticle Associated with Ceftriaxone and Extract of Schinopsis Brasiliensis Engler against Multiresistant Enterobacteria"

_pharmaceutics, 2020, doi:10.3390/pharmaceutics12080695_

Round 1

Reviewer 1 Report

Paper is focused on the investigations on the polymeric nanoparticles based on chitosan and hydroxypropylcellulose and containing selected plant extract and active pharmaceutical ingredient (API). Formed systems were considered as systems for application in therapies against multiresistant bacteria. Paper has been prepared properly. Well-constructed Introduction of the manuscript is followed by well-designed experimental part where main attention is paid to the assessment of antimicrobial activity of systems received. The whole is worth considering for publication, but some changes are recommended. They have been presented in more detail below.

  • Title of the paper is inadequate. Obtained materials are analyzed in viewpoint of their application in treatment against selected bacteria and not for treatment of these microorganisms therefore the title of the paper should be changed.
  • Section 3.: Authors mentioned about the analysis of the quality of the plant drug used for the synthesis. This analysis should be described in few sentences.
  • Section 2.4. presenting the methodology of the polymeric nanoparticles development needs to be supplemented with some information. For example, there is no information concerning the temperature of this process. Next, the information on the method of the pH adjusting to 5.5 has also not been provided. Additionally, what was the form in which API was used during the synthesis? (i.e. as a powder or as a solution/suspension)
  • Section 3.3.: term „nano” refers to the materials whose at least one dimension is within the range 1 – 100 nm therefore the use of this term in reference to particles with sizes between 150 – 500 nm (308 line) is inadequate.
  • Figure : there are some bands on presented spectra which have not been explained (e.g. these ones at approx. 600 – 700 cm-1). Additionally, spectra presented in Figure 6 A) and 6 B) should be presented at the same graph to allow to better compare the differences between samples with and without the extract and API.

Section References should be prepared according to the requirements of the Journal, i.e. with abbreviated journal titles. Next, paper should be re-checked grammatically

Author Response

We would like to thank the Reviewer for their valuable comments concerning our work. All issues raised in the review process are addressed in the Revised Manuscript and word file attached here. The corrected text is highlighted in yellow and described in "List of Changes in the manuscript text". A detailed statement of the modifications made in the manuscript is given below. The text includes more detail to make it more didactic and clearer for the readers.

Reviewer 2 Report

The manuscript Polymeric nanoparticle associated with ceftriaxone and extract of Schinopsis brasiliensis Engler for the treatment of multiresistant enterobacteria explains the anti-microbial nano-composite against multiresistant enterobacteria. The manuscript is scientifically sound and fits in the context of the journal. I recommend minor revisions as above;

  1. Many paragraphs in the introduction section, the introduction needs to revise thoroughly.
  2. Figure 1 is not clear, enhance the resolution and place in supplementary information as it not adding important information
  3. Why polymeric nanoparticles with API caused ant-bacterial effect need to be discussed
  4. There are many short forms, which make the reader confused, so give the list of all abbreviations after keywords.

Author Response

We would like to thank the Reviewer for their valuable comments concerning our work. All issues raised in the review process are addressed in the Revised Manuscript. The corrected text is highlighted in blue and described in "List of Changes in the manuscript text" attached here. A detailed statement of the modifications made in the manuscript is given below. The text includes more detail to make it more didactic and clearer for the readers.
